# Pro-Tumor Activity of Endogenous Nitric Oxide in Anti-Tumor Photodynamic Therapy: Recently Recognized Bystander Effects

**DOI:** 10.3390/ijms241411559

**Published:** 2023-07-17

**Authors:** Albert W. Girotti, Jerzy Bazak, Witold Korytowski

**Affiliations:** 1Department of Biochemistry, Medical College of Wisconsin, Milwaukee, WI 53226, USA; 2Department of Biophysics, Jagiellonian University, 31-007 Krakow, Poland; jerzy.bazak@uj.edu.pl (J.B.); witold.korytowski@uj.edu.pl (W.K.)

**Keywords:** photodynamic therapy, nitric oxide (NO), iNOS/NO upregulation, pro-tumor NO, bystander effects

## Abstract

Various studies have revealed that several cancer cell types can upregulate inducible nitric oxide synthase (iNOS) and iNOS-derived nitric oxide (NO) after moderate photodynamic treatment (PDT) sensitized by 5-aminolevulinic acid (ALA)-induced protoporphyrin-IX. As will be discussed, the NO signaled cell resistance to photokilling as well as greater growth and migratory aggressiveness of surviving cells. On this basis, it was predicted that diffusible NO from PDT-targeted cells in a tumor might enhance the growth, migration, and invasiveness of non- or poorly PDT-targeted bystander cells. This was tested using a novel approach in which ALA-PDT-targeted cancer cells on a culture dish were initially segregated from non-targeted bystander cells of the same type via impermeable silicone-rimmed rings. Several hours after LED irradiation, the rings were removed, and both cell populations were analyzed in the dark for various responses. After a moderate extent of targeted cell killing (~25%), bystander proliferation and migration were evaluated, and both were found to be significantly enhanced. Enhancement correlated with iNOS/NO upregulation in surviving PDT-targeted cancer cells in the following cell type order: PC3 > MDA-MB-231 > U87 > BLM. If occurring in an actual PDT-challenged tumor, such bystander effects might compromise treatment efficacy by stimulating tumor growth and/or metastatic dissemination. Mitigation of these and other negative NO effects using pharmacologic adjuvants that either inhibit iNOS transcription or enzymatic activity will be discussed.

## 1. Introduction

When cancer cells in any given population are targeted by ionizing radiation, many of them may elude or survive the challenge. However, instead of retaining pre-existing characteristics, these so-called “bystander” cells often exhibit more aggressive behavior in terms of more rapid proliferation, migration, and invasion [1,2,3]. The possibility of off-target bystander effects in conjunction with non-ionizing photodynamic therapy (PDT) for cancer has been recognized for many years, but much less is known about these effects than those induced by ionizing radiation. In considering the limitations for optimal PDT (e.g., irregularities in photosensitizer-delivering tumor vasculature, non-uniform light delivery, light scattering, etc.), the authors of this review recognized that not all cells in a given tumor would be uniformly targeted during a PDT challenge. Non-uniform vascular delivery of the photosensitizing agent could be at least partially responsible for this. Moreover, because of vascular variability, not all tumor cells would be exposed to O_2_ to the same extent. Thus, O_2_ deficiency could also spare some sections of a tumor. From these considerations, we hypothesized [4] that cells experiencing significant PDT stress could send pro-survival/expansion signals to non-stressed counterparts (bystanders). We knew that targeted cells surviving a PDT challenge typically overexpress inducible nitric oxide synthase (iNOS) and that resulting NO stimulates proliferative and migratory aggressiveness [5,6,7,8]. From this knowledge, we postulated that this targeted cell NO, by diffusing to naïve bystander cells, would also make them more aggressive [4]. In this review, we discuss such bystander responses, their negative implications for PDT efficacy, and how the latter might be mitigated by specific pharmacologic interventions. A flow scheme reflecting on the rational development of this review based on published findings is shown in Figure 1.

## 2. Nitric Oxide and Its Relevance to Cancer

Nitric oxide is a short-lived free radical molecule (τ < 2 s in H_2_O) that diffuses freely on its own in aqueous media and, like O_2_, tends to partition freely into hydrophobic regions of cells, e.g., mitochondrial and lysosomal membranes [9,10,11]. Naturally occurring NO is produced by enzymes of the nitric oxide synthase (NOS) family, including the neuronal (nNOS/NOS1), endothelial (eNOS/NOS3), and inducible (iNOS/NOS2) isoforms [12]. All three enzymes catalyze the 5-electron oxidation of L-arginine to L-citrulline, and NO, NADPH, and O_2_ serve as co-substrates [12]. Once formed, NO can react with abundant O_2_ to give nitrous anhydride or dinitrogen trioxide (N_2_O_3_); this is one important reason for NO’s short lifetime. NO is pleiotropic in nature and is involved in many different normo-physiologic as well as patho-physiologic processes. Because of its instability, NO must be continuously generated to reach biologically meaningful steady-state levels. For example, eNOS-generated NO in the 1–10 nM range stimulates cyclic-GMP formation in vascular smooth muscle, leading to blood pressure reduction via vasodilation. However, iNOS-derived NO at much higher levels (1 µM or greater), as produced by vascular macrophages in response to infection, is cytotoxic and potentially oncogenic if it causes DNA damage [9,11]. In addition to relaxing vascular smooth muscle, low-level NO can act as a cytoprotective antioxidant by intercepting free radical intermediates (oxyl and peroxyl radicals) that arise during propagative (chain) lipid peroxidation in membranes and lipoproteins [13,14,15]. Thus, NO in the micromolar range tends to be cytotoxic and potentially mutagenic, whereas, in the low-to-medium nanomolar range, it can support survival as well as proliferative and migratory aggressiveness of tumor cells. NO-mediated bystander effects induced by anti-tumor PDT are one example of this acquired aggressiveness, as will be discussed in subsequent sections. Figure 2 illustrates the dichotomy of effects that local steady-state NO concentration can have on tumor cells.

Many cancer cells, including those derived from human breast, prostate, colon, and brain tumors, express significant constitutive levels of iNOS/NO, which are often implicated in tumor persistence and progression [16,17,18]. Knockdown of pre-existing iNOS using siRNA or shRNA methodology has been shown to attenuate the growth and expansion of malignancies [16,17,18], thereby substantiating iNOS/NO’s tumor-supporting role. iNOS level in resected tumors from cancer patients is now considered a reliable prognostic indicator; patients with the highest levels are given the poorest survival chances [19]. Although constitutive or pre-existing iNOS may provide a survival/growth advantage in some tumors, the level of NO produced might still be limiting. One approach for investigating this is to determine whether low-dose NO from a chemical NO donor might stimulate cancer cell proliferation or resistance to therapeutic challenges. In one early example, it was shown that the chemical donor of NO, spermine-NONOate (SPNO), in sub-toxic doses dramatically increased the resistance of human breast cancer COH-BR1 cells to oxidative killing by photodynamic stress [20]. Each of the following was shown to contribute to this response: (a) inhibition of pro-apoptotic JNK and p38α activation; (b) inhibition of pro-apoptotic Bax and Bid expression; and (c) inhibition of anti-apoptotic Bcl-xL down-regulation [20]. Thus, these pro-survival signaling effects of NO were well-coordinated, i.e., they inhibited apoptosis promoters on the one hand while stimulating apoptosis inhibitors on the other. Although not identified in this study [20], it is likely that these NO effects occurred via S-nitrosation of the indicated effector proteins or associated regulators. There is also evidence that wild-type tumor suppressor p53 can either block iNOS transcription or inactivate iNOS by binding to it [21,22]. On the other hand, it has been reported that NO can reduce p53’s pro-apoptotic activity by chemically modifying it and altering its conformation [23]. Interestingly, human carcinoma cells with dysfunctional (mutated) p53 were found to express higher levels of iNOS than wild-type controls, resulting in more rapid proliferation and upregulation of angiogenic factors [24]. As illustrated by these and many other examples, it is now clear that constitutive low-level endogenous iNOS/NO can support the persistence and progression of a variety of human cancers. As we now realize and will discuss next, these factors can be dramatically enhanced in cancer cells that can resist and survive an oxidative photodynamic challenge.

**Figure 2 ijms-24-11559-f002:**
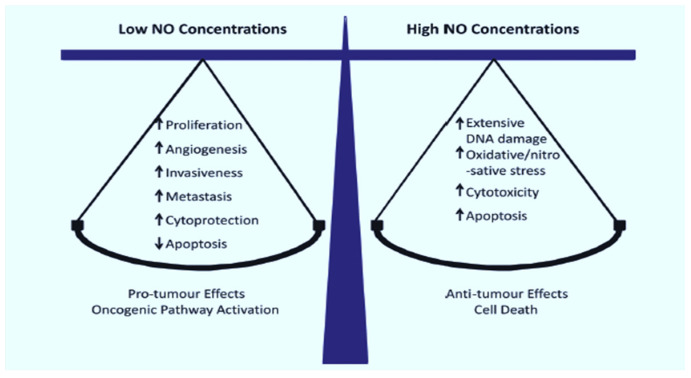
Concentration-dependent effects of NO on cancer cells. Adapted from ref. [19].

## 3. iNOS/NO Antagonism to Anti-Tumor Photodynamic Therapy

Anti-tumor PDT is a unique, minimally invasive therapy involving a photosensitizing agent (PS), molecular oxygen, and PS-exciting light in the far visible-to-near infrared wavelength range [25,26]. PDT is based on cytotoxic photochemical reactions that occur at/near PS localizing sites, i.e., tumor cells per se, or proximal vascular cells, e.g., endothelial cells or fibroblasts. These reactions are mediated by PDT-generated reactive oxygen species (ROS), the most prominent of which is singlet molecular oxygen (^1^O_2_) [26]. Most PSs are innocuous until photoactivated, and PDT has few (if any) negative side effects, unlike chemo- or radiotherapy [26]. The PS can be administered as such, a prime example being Photofrin^®^ (oligomeric hematoporphyrin), the first PS to receive FDA approval for clinical PDT [25]. Alternatively, a pro-sensitizer like 5-aminolevulinic acid (ALA) can be used [27,28]. Upon entering tumor cells via an amino acid receptor, ALA is metabolized via the heme biosynthetic pathway to the functional PS, protoporphyrin IX (PpIX) [27,28]. This pathway is typically more active in rapidly dividing malignant cells than normal counterparts, which increases the selectivity for the former in this approach. PpIX initially accumulates in mitochondria, making these organelles the most prominent early targets of ALA-PDT damage. In addition to high tumor cell selectivity, another advantage of this approach is that the fluorescence of ALA-induced PpIX under low light intensity can be used for fluorescence-guided tumor surgery (FGS) [29]. This approach is now widely used in order to minimize the excision of normal tissue around the tumor area. FGS is often followed up by PDT at higher light doses to eliminate any tumor tissue that is inaccessible to FGS. 

Many cancers exhibit an innate or acquired resistance to various types of chemotherapy or radiotherapy [30], and it is now clear that resistance mechanisms also exist for PDT and that NO is often involved. Some key research findings in support of this are listed in Table 1. How endogenous NO might affect PDT efficacy was first investigated circa 25 years ago by two groups using various mouse-borne syngeneic tumors sensitized with Photofrin^®^ [31,32]. They found that PDT-induced tumor suppression could be much improved when a non-specific inhibitor of NOS activity (e.g., L-NNA or L-NAME) was present during irradiation. Tumors with the highest constitutive NO output exhibited the greatest sensitivity to NOS inhibition [32]. More recent work with ALA/light-treated tumors produced similar findings [33]. The mechanistic reasoning for all these early findings was that vasodilation caused by low-level NO was counteracting PDT’s anti-tumor vasoconstrictive effects [31,32,33]. However, at least three questions remained unanswered: (i) whether constitutive or basal NOS/NO is sufficient for hyper-resistance; (ii) whether stress-induced NOS/NO upregulation is necessary; and (iii) which of the three NOS isoforms is most important. In addressing these questions, Bhowmick et al. [5,6,7] found that the iNOS isoform was the principal source of NO involved in PDT resistance. They referred to this particular PDT as a photodynamic “treatment” rather than “therapy” because the experiments were carried out in vitro using cultured human breast and prostate cancer cells. Importantly, the observed hyper-resistance was found to be mainly due to PDT-upregulated iNOS rather than pre-existing/constitutive enzymes. At the time, such a finding about iNOS/NO was unprecedented for any type of cancer therapy. ALA-induced PpIX was the PS used in the above studies [5,6,7], so any PpIX export via the ABCG2 transporter [34] could have also imparted some resistance. In this case, however, no photodynamic stress would have been involved, in contrast to NO-induced resistance. Recognition of iNOS/NO involvement in PDT resistance was based on findings such as the following: (i) strong mitigation by a NO scavenger (cPTIO) or by specific inhibitors of iNOS activity (1400 W, GW274150), apoptotic cell death increasing in response; (ii) prevention by siRNA-based iNOS knockdown, apoptosis being elevated in response; (iii) substantial “rescue” of iNOS-knockdown cells with spermine-NONOate (SPNO), a chemical NO donor [5,6]. 

In 2017, the above in vitro findings were replicated and substantiated at the in vivo level. Immunodeficient (SCID) female mice bearing human breast MDA-MB-231 tumor xenografts were sensitized with ALA-induced PpIX and irradiated with 630 nm LED light [35]. This caused a significant slowdown in tumor growth over a 12-day post-irradiation period. Administration of 1400 W or GW274150 during this period slowed growth even further, consistent with iNOS/NO-dependent resistance [35]. Immunoblot and NO analyses on post-PDT tumor samples revealed a progressive increase in iNOS expression and NO output (each reaching 5–6-fold over starting level at 6 h post-PDT). A light-only control was unresponsive to 1400 W, indicating that overexpressed, but not basal iNOS/NO, was promoting tumor growth/persistence after a PDT challenge [35]. Consistently, anti-apoptotic Bcl-xL, Survivin, and S100A4 underwent 1400 W-inhibitable upregulation after the challenge, whereas pro-apoptotic Bax was down-regulated [35]. Collectively, this in vivo evidence was the first of its kind for an anti-tumor therapy that is opposed by the iNOS/NO it induces.

In addition to resisting eradication, certain tumor cells that survive PDT stress have been found to exhibit more aggressive behavior than non-stressed controls. For example, when human prostate cancer PC3 cells remaining alive (attached) after an ALA/light challenge were continuously monitored beyond 24 h, a strikingly (~3-fold) increase in proliferation rate was observed relative to dark (ALA-only) controls [8]. This growth spurt was abolished by 1400 W or cPTIO, signifying iNOS/NO involvement. Of added significance was the discovery that surviving PC3 cells were more motile, as manifested by increased migration and invasion rates, with iNOS/NO again playing a key driving role [8]. Enhanced iNOS/NO-mediated resistance, as well as growth and migratory aggressiveness, has also been observed for human glioblastoma U87 cells that can withstand PDT stress [36]. For example, in addition to resisting mitochondria-initiated apoptosis, PDT-surviving U87 cells underwent a strong growth and invasiveness spurt, which, as with PC3 cells, was 1400 W-inhibitable, demonstrating iNOS/NO-dependency [36]. Importantly, this dependency was on stress-upregulated iNOS (>3-fold at 24 h post-PDT) rather than background iNOS, which was not significantly affected by 1400 W. Matrix metalloproteinases (MMPs) are known to play a key role in cancer cell invasiveness and metastasis. ALA/light stress markedly increased MMP-9 activity in ALA/light-stressed PC3 and U87 cells [8,36]. In each case, inhibition of this activity elevation by 1400 W signified iNOS/NO dependency [8,36]. Moreover, expression of TMP-1, which inhibits MMP-9, was progressively reduced, demonstrating cooperative responses that promoted surviving-cell migration/invasion [8]. 

Considerable evidence about the signaling events that drive iNOS upregulation under PDT stress has also been obtained. In at least two human cancer lines, breast COH-BR1 and glioblastoma U87, iNOS induction required initial activation of transcription factor NF-κB. For ALA/light-challenged COH-BR1 cells, Bay11-7082, an inhibitor of phosphorylation and proteasomal removal of NF-κB regulatory subunit, IκB: (i) prevented translocation of NF-κB subunit p65 to the nucleus to initiate transcription, (ii) suppressed iNOS upregulation after the photo challenge, and (iii) increased the extent of cell photokilling via apoptosis [37]. Key signaling events upstream of NF-κB have also been identified. For example, in ALA/light-challenged COH-BR1 and U87 cells, the pro-tumor kinases PI3K and Akt were rapidly hyper-activated via phosphorylation [38]. PI3K inhibitors such as Wortmannin and LY294002 strongly suppressed Akt activation, NF-κB activation, and iNOS upregulation while stimulating apoptosis [37,38]. Thus, a PI3K → Akt → NF-κB → iNOS signaling sequence was in operation. For COH-BR1 cells, ODQ, an inhibitor of soluble guanylate cyclase (sGC), failed to enhance photo stress-induced apoptosis, thereby arguing against NO/sGC/cGMP-mediated activation of protein kinase-G, a cancer cell pro-survival/expansion effector. However, pro-apoptotic Bax was upregulated and anti-apoptotic Bcl-xL was downregulated, 1400 W or cPTIO accentuating these responses, consistent with NO playing a strong pro-survival/anti-apoptotic role [38]. Additional evidence on upstream signaling revealed that for ALA/light-challenged U87 cells, transacetylase p300 underwent greater: (i) Akt-dependent activation and (ii) interaction with NF-κB subunit p65, which exhibited hyper-acetylation at lysine-310 (acK310) [38]. Moreover, photostressed U87 cells exhibited greater inactivating disulfide bond formation in tumor suppressor PTEN. This would have favored the activation of p300 and Akt, leading to greater iNOS transcription via the interaction of p65-acK310 with bromodomain protein Brd4 [39]. In addition to this, deacetylase Sir1 was down-regulated by photo stress, which would have favored the observed increase in p65-acK310 level and hence greater iNOS transcription [38]. These findings with two different human cancer lines, COH-BR1 and U87, provided important mechanistic insights on how these, and presumably other cancer cells, may not only resist a PDT challenge but respond more aggressively if capable of withstanding it. When obtained, these findings [37,38,39] were unique in the overall field of anti-tumor PDT biology/biochemistry. 

Certain chemo and radiotherapeutic strategies based on oxidative stress are also known to be antagonized by iNOS/NO. For chemotherapy, cisplatin (CDDP) is the most widely recognized drug to elicit antagonism. For example, one study involving CDDP-treated melanoma cells showed that two key pro-apoptotic proteins were inhibited by S-nitrosation: caspase-3 and prolyl hydroxylate-2 (PHD2), which otherwise targets pro-survival HIF-1α for proteasomal degradation [40]. Another example pertains to tumor-associated macrophages (TAMS), specifically M2-TAMS, which play an important role in tumor survival, expansion, and drug resistance. A recent study showed that iNOS-derived NO from human and murine glioma M2-TAMs induced CDDP resistance by inhibiting acid sphingomyelinase (A-SMase), which otherwise stimulates apoptosis via death receptor CD95 [41]. Interestingly, the anti-therapeutic NO in this case, unlike that described above for PDT models [35,36,37,38,39], was not emitted by stressed tumor cells per se but rather by TAMs in the tumor microenvironment. In regard to ionizing radiation, a recent study showed that immortalized as well as patient-derived glioblastoma cells responded to fractionated radiation (2 Gy of γ-rays over three days) by expanding stem-like cell (GSC) populations with upregulated iNOS/NO [42]. Post-radiation GSC expansion was resistant to higher dose radiation (10 Gy) and CDDP treatment. However, prior iNOS knockdown prevented all responses, thus confirming iNOS/NO instigation. As another example, pancreatic ductal adenocarcinoma (PDAC), like glioblastoma, is difficult to treat and highly lethal. Low-level iNOS-derived NO was found to play a key role in PDAC persistence and progression after radiation exposure [43]. Another highly relevant study showed that iNOS/NO was strongly upregulated in PDAC-associated fibroblasts (CAFs) after moderate exposure to ionizing radiation [44]. When treated with these CAFs, PANC cells exhibited a striking upregulation of iNOS/NO, which stimulated cell proliferation and migration [44]. Enhancing their significance, these findings were subsequently validated at the in vivo level using orthotopic PDAC tumors in mice [44]. 

**Table 1 ijms-24-11559-t001:** Examples of iNOS/NO-stimulated anti-PDT/pro-tumor effects.

Photosensitizer	System Studied	Key Findings	Reference
Photofrin	Mouse-borne syngeneic tumors (RIF, SCCVII, EMT6	Highest NO-producing tumors—most resistant	[32]
ALA-induced PpIX	Human prostate PC3 cells irradiated in vitro	PDT-induced iNOS/NO; PDT resistance	[7]
ALA-induced PpIX	Human prostate PC3 cells irradiated in vitro	Accelerated growth/migration of cells surviving PDT	[8]
ALA-induced PpIX	MDA-MB-231 tumor xenografts in SCID mice	Stimulation of tumor growth by upregulated iNOS/NO	[35]
ALA-induced PpIX	U87 glioblastoma cells irradiated in vitro	BET inhibitor suppression of cell hyper-aggressiveness after PDT	[39]
3-THPP, PpIX	MDCXII cells in vitro	Discovery of PDT-induced bystander effect (undefined, but likely due to NO)	[45]
Deuteroporphyrin	WTK1 lymphoblastoid cells in vitro	Cytotoxic bystander effect (poorly defined mechanism, but possible iNOS/NO role)	[46]
ALA-induced PpIX	Prostate PC3 cells irradiated in vitro	Stimulation of bystander cell growth/migration by targeted cell iNOS/NO	[47]

## 4. Bystander Effects of PDT-Upregulated iNOS/NO

When exposed to physical or chemical perturbations, cancer cells often send stress signals to unperturbed neighboring cells (bystanders). This may cause the latter to respond in diverse ways, ranging from viability loss to stimulated migration and proliferation, depending on signal intensity. This phenomenon, referred to as a “bystander effect”, was first recognized as an unexpected side effect of ionizing radiation [1,2,3]. When such radiation (e.g., X-rays, γ-rays, α-particles) is used for anti-tumor therapy, not all cells in any given solid tumor will be targeted uniformly due to many technical limitations [3]. However, well-exposed cells are known to elicit a variety of stress responses in non- or minimally exposed bystanders, ranging from DNA damage, faulty damage repair, and apoptotic cell death to faster proliferation and migration [1,2,3]. Stress signals can be sent by at least two different mechanisms: (i) via gap junctions between cells or (ii) via the extracellular medium, i.e., without actual physical contact between cells [2,3]. The latter mechanism is illustrated schematically in Figure 3A. 

For ionizing radiation, a variety of signaling molecules capable of traversing aqueous media from targeted to bystander cells have been identified, e.g., (i) cytokines such as TGF-β and TNF [48,49], (ii) ROS such as H_2_O_2_ [50], and (iii) NO or NO-derived species like nitrous anhydride (N_2_O_3_) [51,52]. Like H_2_O_2_, continuously generated NO diffuses rapidly in aqueous media, but unlike H_2_O_2_, NO can partition into low-polarity lipoprotein or membrane sites. In addition, NO is not susceptible to any known enzymatic scavenging, which distinguishes it from H_2_O_2_; Thus, enzymatic scavenging could not impose limitations on NO’s sphere of activity, including potential bystander activity. 

The bystander effects of non-ionizing PDT were first described over twenty years ago [45], but intercellular signaling mechanisms were not defined in rigorous biochemical terms. Subsequent studies with cancer cells involved (i) a plasma membrane-bound PS with trans-well inserts to separate targeted from bystander cells [46] or (ii) time-lapse fluorescence microscopy to monitor two non-contacting cell groups on a culture dish after either one was PDT-stressed [53]. The only endpoint monitored in these studies was the ROS-induced death of targeted or bystander cells. There was no consideration of a possible iNOS/NO role in the observed responses or whether any photo stress-surviving cells might divide or migrate more aggressively. 

These questions were directly addressed by Bazak et al. [47] about six years ago. The investigators devised a novel approach based on silicone-rimmed rings for initially separating photodynamically-targeted cancer cells from non-targeted bystanders. Figure 3B illustrates a typical experimental arrangement with two rings in place on a large culture dish. Human prostate cancer PC3 cells were used in initial studies [47]. Target cells (outside rings) were sensitized with ALA-induced PpIX, while PC3 bystanders (inside rings) saw no ALA. The cells were then exposed to ~1 J/cm^2^ of LED irradiation. After a suitable dark delay, e.g., 2 h, the rings were carefully removed, allowing diffusible stress-upregulated molecules like NO to flow from targeted to bystander zones. As anticipated from previous findings [5,6,7,8], surviving targeted cells exhibited a very robust upregulation of iNOS, which reached ~12-fold the control level 4 h after irradiation and remained there for at least another 20 h (Figure 4A). Importantly, iNOS was also upregulated in non-stressed bystander cells, albeit more slowly and less extensively, reaching four to five times its starting level after 24 h (Figure 4B) [47]. Target cell controls (ALA alone or light alone) showed no significant increase in iNOS above background for either of the two cell populations. Unlike iNOS, nNOS and eNOS exhibited no change from their low initial levels in PC3 cells after the photodynamic challenge. Thus, iNOS appeared to be unique in this capacity, as substantiated by several other cancer lines [4,35,36]. Moreover, iNOS knockdown in target PC3 cells markedly suppressed the enzyme’s induction in bystanders, suggesting that iNOS-derived NO from the former was essential [47]. This was confirmed by showing that the NO trap, cPTIO, strongly inhibited bystander iNOS induction, thus revealing that diffusible NO generated continuously by targeted cells was the active mediator. In agreement with this, a conditioned medium from these cells had no significant effect [47], thus indicating that relatively long-lived agents such as hydroperoxides or cytokines could not have been involved. Also ruled out were relatively stable byproducts of NO, such as N_2_O_3_ and nitrite (NO_2_^−^). In addition to iNOS, at least three other proteins were either upregulated or hyperactivated in targeted and bystander cells. Cyclooxygenase-2 (COX-2), which is expressed in many different tumors and plays a pro-survival (anti-apoptotic) role, was upregulated in ALA/light-targeted and bystander cells, though more slowly in the latter, after irradiation [47]. Both cell responses were strongly attenuated by the NO scavenger, cPTIO, implying NO-dependency. In contrast to COX-2, protein kinases Akt and ERK-1/2 were not overexpressed but became more enzymatically active, as indicated by greater phosphorylation in Western blot bands [47]. Phosphorylation of Akt and ERK-1/2 was nullified by the NO trap, cPTIO, again signifying NO-dependency. In accordance with these responses, a striking increase in proliferative and migratory aggressiveness was observed for bystander PC3 cells, and, once again, this was dependent on stress-upregulated NO released from targeted cells [47]. For the experiment represented in Figure 5A, bystander cells were dividing at least 50% faster than controls at 39 h after irradiation. Likewise, at 21 h after irradiation, bystanders were migrating at least 30% faster than their control cells (Figure 5B). Thus, bystander cells were replicating the enhanced aggressiveness exhibited by cells that remained alive and active in the PDT-targeted compartment. 

In follow-up work, Bazak et al. [54] compared the bystander behavior of prostate PC3 cells with that of three other human cancer lines: melanoma BLM, glioblastoma U87, and breast MDA-MB-231. ALA treatment was the same for all four lines, but subsequent light fluences were varied such that a uniform targeted cell kill was attained for all, i.e., ~25% at 24 h after irradiation [54]. Targeted cell iNOS levels were assessed by immunoblotting at various post-irradiation times up to 24 h. Enhanced bystander cell proliferation or migration was tracked as a function of maximal iNOS upregulation. As shown in Figure 6A, the greater proliferation rate of bystander cells increased exponentially with maximal targeted iNOS induction for the four cancer lines examined, with BLM cells exhibiting the smallest change and PC3 cells the greatest. A similar exponential increase in elevated migration rate was observed for the four bystander lines (Figure 6B); PC3 cells with maximal iNOS induction showed, again, the maximal effect. Thus, the extent to which the growth and migration of bystander cells escalated depended on the degree to which iNOS/NO was boosted in PDT-targeted cells [54]. As initially indicated, surviving targeted cells also exhibited greater growth/migration aggressiveness (Section 3); however, this was self-promoted, i.e., directly due to upregulated iNOS/NO in the targeted population. Since NO from targeted cells stimulated iNOS upregulation in bystanders (Figure 4B), it appears that a “feed-forward field effect” was in operation whereby the signaling effects of NO from PDT-challenged cells were “broadcasted” to non-challenged bystanders, thereby disseminating pro-growth/migration stimuli. When reported, these observations [4,47,54] were unprecedented in the context of PDT. Their significance was enhanced by another group’s recent studies, which described differences in bystander behavior for prostate cancer cells of varying malignancy grades [55].

The mechanism by which elevated NO diffusing from targeted to bystander cells stimulates iNOS expression in the latter (see Figure 2) is unknown. Recall that upregulation of targeted-cell iNOS requires PI3K and Akt activation followed by NF-κB activation with nuclear transfer and acetylation of p65 (Section 3). However, this sequence of events was set off by photooxidative stress-signaling in targeted cells [47,54], whereas no oxidative pressure was imposed on naïve bystanders. Cytokines such as interferon-γ (INF-γ), interferon-β (INF-β), and tumor necrosis factor-α (TNF-α) are known to stimulate iNOS expression in mammalian cells [56,57]. It is possible that NO continuously emanating from targeted cells somehow activated one or more of these cytokines, which in turn could have stimulated iNOS formation. More comprehensive PDT-based studies are needed in order to deduce the precise mechanism of bystander induction by targeted cell-derived NO. 

As indicated in Section 3, there are many examples of cancer cells that exploit NO to divide and migrate more aggressively after withstanding a chemo- or radiotherapeutic attack. Existing evidence for ionizing- and PDT-induced bystander effects [1,2,3,4] suggests that similar effects might occur during chemotherapeutic treatments based on imposed oxidative stress. In support of this were studies showing that endogenous NO stimulates cisplatin resistance in tumor cells [40,41]. Whether cisplatin-induced bystander effects occurred in those studies was not assessed, but there is a strong likelihood, given that PDT oxidative stress elicits such effects [47,54,55]. Thus, NO-mediated hyper-aggressiveness of bystander cells may not be limited to PDT but could extend to cisplatin-based and other oxidative chemotherapies. We look forward to future studies dealing with such pro-tumor bystander effects and how they might be mitigated. 

## 5. Pharmacologic Suppression of iNOS/NO Anti-Therapeutic Effects

If occurring during clinical PDT (or even chemotherapy or radiotherapy), the iNOS/NO effects described could have negative effects on treatment outcomes, particularly if significant numbers of tumor cells can withstand the challenge or lie in NO-accessible bystander regions. Pharmacologic inhibition of iNOS activity has been advocated for tumors that significantly rely on iNOS-derived NO for the promotion of growth and metastatic dissemination. Activity inhibitors, such as L-NAME and L-NNA, have been tested in in vitro and in vivo model systems [2,4,5,6]. Such inhibitors could be used alone or in combination with PDT treatment. However, these compounds are not specific to iNOS and might also inhibit eNOS or nNOS. Thus, the possibility of compromising normal physiologic functions such as blood pressure regulation and bacterial infections would be a concern. At least two activity inhibitors, L-NIL and GW274150, which are highly specific for iNOS, have been safely tested in clinical trials, although these had no relationship to cancer or cancer therapeutics [58,59]. Such agents in appropriate doses and with sufficiently long post-administration lifetimes might improve the efficacy of PDT for cancer patients, and we look forward to their clinical testing for this purpose. 

Another possibility for mitigating the negative effects of overexpressed iNOS on PDT would be to suppress its expression at the transcriptional level. Recent studies [38,39] have shown that accelerated growth and migration of PDT-surviving glioblastoma cells in vitro could be prevented by JQ1, an inhibitor of the bromo/extra-terminal (BET) domain of Brd4, an epigenetic “reader” protein [60,61]. Brd4, along with the acetylated p65 subunit of NF-κB, was found to be essential for iNOS transcription in these cells [38,39]. At a 100-fold lower concentration than 1400 W or GW274150, JQ1 was much more effective in suppressing post-PDT hyper-aggressiveness than either of these other inhibitors. This was the first known evidence that a BET inhibitor, by blocking iNOS expression, could strongly diminish cancer cell aggressiveness [39]. JQ1 may have also interfered with the transcription of other pro-tumor proteins such as Bcl-xL, MMP-9, and Survivin. If their levels were reduced, however, this might have been an indirect effect of JQ1 on iNOS because NO is known to signal for induction of these proteins [39]. BET inhibitors like JQ1 have been shown to be highly effective against several cancers at the in vitro, animal model, and clinical trial levels. Hence, there is great promise in their being used as adjuvants for anti-tumor PDT. Thus, there is good reason to begin clinical trials on BET inhibitors as PDT adjuvants as soon as possible. 

## 6. Summary and Outlook

There is now substantial evidence that pre-existing and/or overexpressed iNOS/NO can compromise anti-tumor PDT and possibly stimulate disease progression if the extent of tumor eradication is not great enough. Sensitizer or pro-sensitizer uptake by cells in any given tumor is not expected to be uniform throughout, nor is light delivery or O_2_ availability. As a result, some cells will be more heavily stressed by photodynamic action than others, some of which could be relatively unaffected bystanders. Using in vitro model systems, the authors and coworkers have shown that surviving targeted cancer cells strongly upregulate iNOS and that its continuously generated NO promotes cell survival and expansion [62]. Moreover, diffusion of this NO to bystander cells induces iNOS/NO there, stimulating cell growth and migratory aggressiveness. Metastatic dissemination of malignant cells would be a particularly unfortunate outcome of such enhanced aggressiveness. These effects have the overall appearance of a “relay-type” process whereby NO overproduced by targeted cells stimulates NO generation in nearby bystanders, which can then propagate it to other naïve cells; see Figure 7 for an illustration. This is an example of a “feed-forward” process, a term first used for describing the bystander effects of ionizing radiation [1,2,3]. Similar propagation could occur in the targeted cell population. However, it was only through the evaluation of separated cell populations that this phenomenon came to be realized [4,47]. In the genre of radiation biology, this process has also been represented as a “NO feed-forward field effect” [3]. As documented for targeted cells that survive a PDT challenge, enhanced proliferative and migratory/invasive aggressiveness of bystander cells is likely to promote tumor growth and metastatic expansion. These negative side effects could be mitigated by introducing inhibitors of iNOS activity as PDT adjuvants. Two candidates in this regard are L-NIL and GW274150, which have already been tested in clinical trials, although these were not cancer- or PDT-related [58,59]. A more promising approach for suppressing iNOS/NO-enhanced aggressiveness of PDT-surviving or bystander cells would be to administer a BET inhibitor like JQ1. This is supported by the following observation. At a far lower concentration than GW274150, JQ1 was much more effective in inhibiting the accelerated proliferation of PDT-surviving glioblastoma cells [39]. JQ1 should also act as a strong suppressor of the enhanced aggressiveness of post-PDT bystander cells. However, direct experimental evidence for the latter effect has not yet been obtained. On their own, JQ1 and other BET inhibitors are very effective against a variety of malignancies [60,61]. Consequently, administering such inhibitors as PDT adjuvants holds great promise for limiting the negative effects of upregulated iNOS/NO [62]. Thus, we look forward to the first clinical testing of BET inhibitors for this purpose.

## Figures and Tables

**Figure 1 ijms-24-11559-f001:**
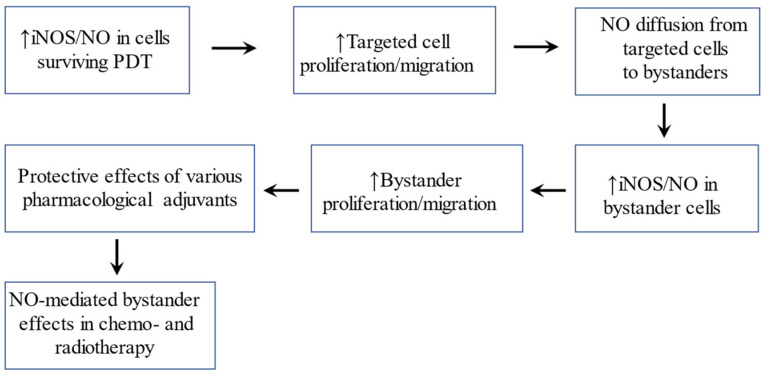
Flow scheme of key background information on which this review is based.

**Figure 3 ijms-24-11559-f003:**
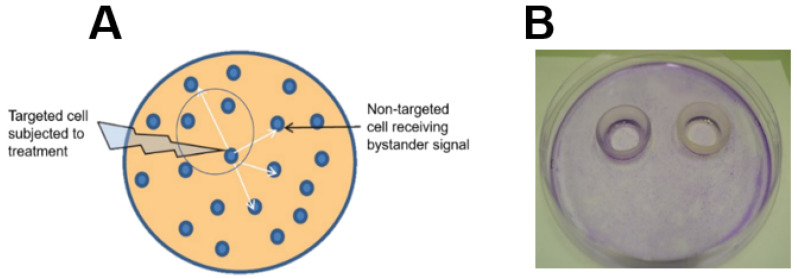
PDT-induced bystander effects and how they can be assessed. Panel (**A**): Schematic showing a population of cancer cells, some of which (targeted) are exposed to a photodynamic challenge while others (bystanders) receive pro-division/migration signals from targeted counterparts. Panel (**B**): View of an experimental setup for initially separating targeted and bystander cell populations on a large (13.5-cm. diam.) culture dish. Two silicone-rimmed rings segregate the two populations, with targeted cells lying outside the rings and bystanders inside. At some point after a challenge, the rings are removed, and the effects of signaling molecules diffusing from outside to inside are examined.

**Figure 4 ijms-24-11559-f004:**
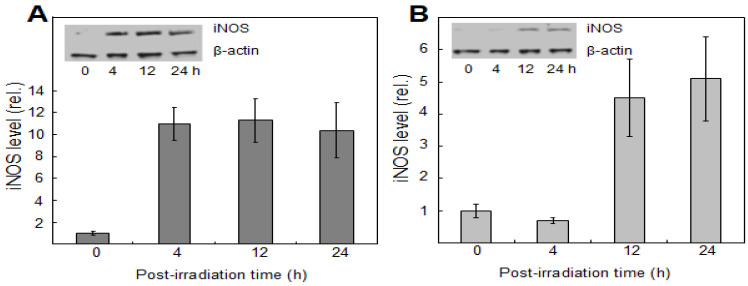
Western blot-assessed iNOS upregulation in photodynamically-targeted PC-3 cells (**A**) and bystander counterparts (**B**). Target cells were dark-incubated with 1 mM ALA for 40 min in serum- and a phenol-red-free RPMI medium. After switching to an ALA-free medium, cells were exposed to ~1 J/cm^2^ of LED light. Non-sensitized bystander cells within two silicone rings apart from targeted cells were irradiated simultaneously. At 2 h post-irradiation, rings were removed, and the medium was switched to 10% serum in RPMI. After the indicated periods of additional dark incubation, targeted and bystander cells were recovered for iNOS and β-actin immunoblot analysis at 30 µg total cell protein per lane. Plotted values indicate iNOS band intensity relative to β-actin and normalized to time 0, means ± SEM (n = 3). No difference was seen between the indicated time 0 values and a target-cell dark control (ALA-only). Adapted from ref. [54].

**Figure 5 ijms-24-11559-f005:**
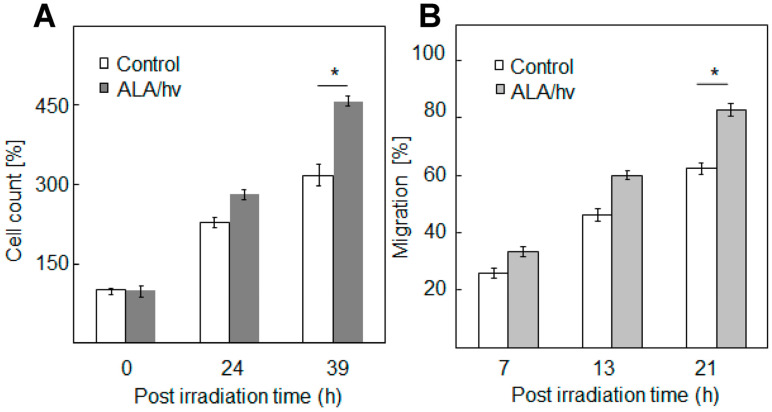
Accelerated proliferation and migration of PC3 bystander cells after exposure to photodynamically-targeted counterparts. Target cells separated from ring-enveloped bystanders were incubated with 1 mM ALA for 40 min, then switched to an ALA-free medium and irradiated as described in Figure 3. A non-ALA-treated control was irradiated alongside. (**A**) Bystander proliferation during post-irradiation incubation as assessed by Image-J analysis; means ± SEM (n = 6), * *p* < 0.05 vs. control. (**B**) Bystander migration determined by gap-closure (“wound-healing”) assay; means ± SEM (n = 5), * *p* < 0.05 vs. control.

**Figure 6 ijms-24-11559-f006:**
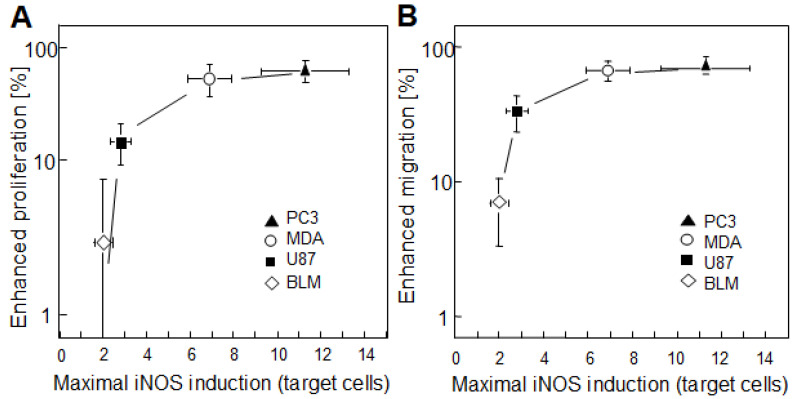
Accelerated proliferation and migration of PC3 bystander cells compared with BLM, U87, and MDA-MD-231 bystander cells as a function of maximal post-irradiation iNOS upregulation in corresponding targeted cells of each type. (**A**) Plot of increased proliferative potency vs. maximal targeted-cell iNOS induction for the different cell lines. (**B**) Plot of increased migration potency vs. maximal targeted-cell iNOS induction for the different lines. All data points are means ± SEM (n = 3–6). Adapted from ref. [54].

**Figure 7 ijms-24-11559-f007:**
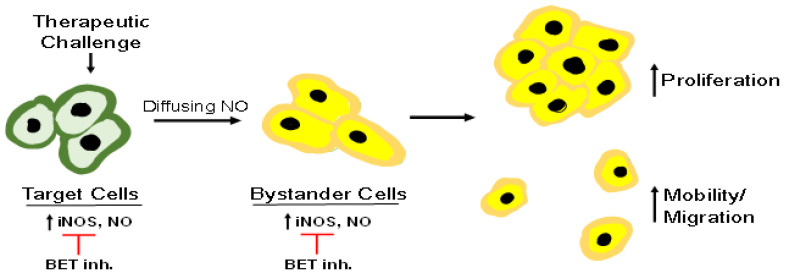
Schematic depicting (a) upregulated iNOS in cancer cells directly targeted by a PDT oxidative challenge, (b) diffusion of the resulting NO to naïve bystander cells, (c) iNOS/NO upregulation in the latter cells, which increases proliferative and migratory aggressiveness, and (d) suppression by BET inhibitors.

## Data Availability

Not applicable.

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
