# Peer review of "Pro-Tumor Activity of Endogenous Nitric Oxide in Anti-Tumor Photodynamic Therapy: Recently Recognized Bystander Effects"

_ijms, 2023, doi:10.3390/ijms241411559_

Round 1

Reviewer 1 Report

I have carefully read the manuscript of Albert W. Girotti et. al. titled “PRO-TUMOR ACTIVITY OF ENDOGENOUS NITRIC OXIDE IN ANTI-TUMOR PHOTODYNAMIC THERAPY: RECENTLY RECOGNIZED BYSTANDER EFFECTS”.

In my opinion, the work was done at a high scientific and technical level, written in a fairly accessible language.

However, there are a number of minor deficiencies in the work that must be corrected before publication in the IJMS journal.

 Minor remarks:

1) In table 1, you need to fix the text formatting

2) Figure 4 is too "stretched" horizontally

3) In the conclusion, you need to remove the unnecessary paragraph break.

4) The terms in vivo, in vitro, et. al. and similar ones should be written in italics

Author Response

Response to Reviewer 1:

1) Thanks, we have now formatted Table 1 properly in revised manuscript.

2)  Fig. 4 too "stretched": We need to use this figure as presented - otherwise Western blot bands are much distorted if we compact it.

3) Conclusion - remove unnecessary paragraph break: OK we have done this.

4) in vivo, in vitro, et al: OK, italics were used for all these terms. 

Reviewer 2 Report

In this work, Girotti and co-workers review the literature regarding the pro-tumor activity of endogenous nitric oxide in photodynamic therapy of cancer. More specifically, they have focused on the recently recognized bystander effects, which might lead to enhanced growth, migration, and invasiveness of non-PTD-targeted, or poorly targeted, bystander cells. Given that these effects might compromise the efficiency of this therapy, this topic is of great relevance.

This is appears to be a comprehensive overview of the topic, and is well-written and easy to follow. In my opinion, it should be useful to both specialists and non-specialists, and of interest for the readership of this journal. I have just one minor comment:

-Lines 435-440: there is mention to conclusions obtained from results that still didn't undergo peer-review nor are available to the reader of this work. So, I do not think that it is correct to mention them, and these conclusions should be removed from the text.

Author Response

Previous lines 435-440: Yes, we agree and have removed this reference to preliminary (unpublished) work. This section has been revised accordingly without mentioning this preliminary work and is highlighted in yellow.